# Synthesis and Biological Evaluation of Quercetagetin Derivatives as the Inhibitors of Mcl-1 and Bcl-2 Against Leukemia

**DOI:** 10.3390/ijms26062727

**Published:** 2025-03-18

**Authors:** Kang Li, Xiaomei Ge, Wei Liu, Lei Huang, Xinye Lv, Yuhui Tang, Zhehao He, Yingxue Yang, Miaofen Chen, Jianguo Zeng, Pi Cheng

**Affiliations:** 1Hunan Key Laboratory of Traditional Chinese Veterinary Medicine, Hunan Agricultural University, Changsha 410128, China; likang_9824@163.com (K.L.); gxm020428@163.com (X.G.); liuwei618@hunau.edu.cn (W.L.); hl1207040882@gmail.com (L.H.); lxyhhh2333@163.com (X.L.); tyhss666@163.com (Y.T.); 13164156185@163.com (Z.H.); yyx085279@163.com (Y.Y.); miaofen_chen@hunau.edu.cn (M.C.); 2Chinese Medicinal Materials Breeding Innovation Center of Yuelushan Laboratory, Changsha 410128, China

**Keywords:** Bcl-2, Mcl-1, apoptosis, quercetagetin derivatives, leukemia

## Abstract

B-cell lymphoma-2 (Bcl-2) family proteins are fundamental regulators of intrinsic cell apoptosis, and overexpression of apoptotic proteins (Bcl-2 and Mcl-1) is a characteristic of many haematological malignancies. Thus, it is necessary to discover novel inhibitors to treat leukemia. In the current study, we synthesized a series of quercetagetin derivatives (compounds **2a**–**2t**, **3a**–**3j** and **4a**–**4g**) and evaluated their anticancer activities on four leukemia cells (U937, K562, K562R and KG-1). Among those synthesized derivatives, compounds **2a** exhibited the best antiproliferative activity (IC_50_ = 0.276, 0.159, 0.312 and 0.271 µM to U937, K562, K562R and KG-1, respectively). In addition, **2a** induced apoptosis in K562 and markedly arrested the cell cycle G2/M phase of K562. The Western blot assay showed that **2a** is a potential inhibitor that can effectively suppress the expression of Bcl-2 and Mcl-1. The molecular docking study predicted that **2a** had firm interactions with the active pockets of Bcl-2 and Mcl-1. Finally, in silico pharmacokinetic evaluation of **2a** indicated its potential as an anti-leukemia drug lead in the future.

## 1. Introduction

Leukemia is a hematological malignant tumor and can be divided into four major categories, acute lymphoid leukemia (ALL), chronic lymphocytic leukemia (CLL), acute myeloid leukemia (AML) and chronic myelogenous leukemia (CML) [1], according to the process rate of disease and the origin of predominant cell type. Leukemia mainly occurs in children and the elderly, and it is the leading cause of cancer-related death among children and comprises about 30% of all childhood cancers [2]. Although many strides have been made in curing leukemia, due to drug resistance and poor prognosis, there are still urgent requirements to find more novel drugs to provide efficient therapy [3,4].

B-cell lymphoma-2 (Bcl-2) family proteins play a crucial role in moderating cell intrinsic apoptotic pathways by the interaction between anti-apoptotic and pro-apoptotic members. They comprise a pro-survival group (Bcl-2, Bcl-xL, Bcl-W, Bcl-A1 and myeloid cell leukemia 1 (Mcl-1)), effector proteins (BAK, BAX) and BH3-only pro-apoptotic initiators (Bim, Bid, Bad, PUMA, NOXA) [5]. Overexpression of anti-apoptotic proteins Bcl-2 [6,7,8,9,10,11], Bcl-XL [12,13] and Mcl-1 [14,15,16] cause avoidance of apoptosis and drug resistance and feature as the hallmarks of many hematological malignancies. Therefore, there is wide acknowledgement that program cell death is a mechanism for tumor suppression, and it is pivotal to develop specific Bcl-2 family inhibitors.

Over the past few years, a lot of Mcl-1 inhibitors [17,18,19] (Figure 1) with potent efficacy in vivo have been exploited, and some of these compounds have been tested clinic trials. However, because of the intrinsic toxicity of those compounds and the multi-physiological functions of Mcl-1, no Mcl-1 inhibitor has been approved yet.

Natural products with remarkable bioactivities are the main source of lead drugs, and they commonly occur in plants. Flavonoids (Figure 2) are a type of phytochemicals with a great variety of pharmacological activities, antioxidant activity [20,21], cardiovascular disease protection [22,23], vascular fragility [24] and anti-inflammatory activity [25,26]. Quercetin and its derivatives have been reported to demonstrate excellent anticancer activity and modulate the expression of anti-apoptotic proteins (Mcl-1 and Bcl-2) to induce apoptosis [27,28,29,30,31,32,33,34,35,36]. In light of recent literature, in this paper, our team chose quercetagetin, a flavonoid extracted from *Tagetes erecta L*. with a similar structure to quercetin, as starting material, designed and synthesized a series of its derivatives and evaluated their antiproliferative activity against four leukemia cell lines. The inhibitory activities of compound **2a** against the expression of Mcl-1, Bcl-2 and other oncoproteins were assayed by Western blot and the effects of compound **2a** on the apoptosis and cell cycle of K562 were tested by flow cytometry assay. Moreover, the molecular docking study predicts the bonding affinity and the interactions among compound **2a**, Mcl-1 and Bcl-2. The drug-like properties of compound **2a** have been analyzed in in silico pharmacokinetic studies.

## 2. Results

### 2.1. Chemistry

The preparation of compounds **2a**–**2t**, **3a**–**3j** and **4a**–**4g** (Figure 1).

In part A, compound **2a** and **3a** were synthesized by reaction between compound **1** (quercetagetin) and CH_3_I under the presence of K_2_CO_3_ in DMF. In part B, under the presence of K_2_CO_3_, compound **2a** or **3a** was reacted with BrCH_2_COOC_2_H_5_ in THF at 60 °C to obtain compound **2b** or **3b**, respectively, and subsequent ester hydrolysis with 1 N NaOH in acetone was carried out to obtain phenyloxyacetic acid derivative intermediates, which further reacted with amino acid ester and ester hydrolysis was carried out again to obtain compounds **2c**–**2j** and **3c**–**3j**. In part C, the compound **2a** was treated with different halogenated alkanes or alkenes in the presence of K_2_CO_3_ in DMF to gain compound **2k**–**2t**. In part D, compound **1** was reacted with halogenated alkanes or alkenes in the presence of K_2_CO_3_ in DMF at 60 °C to obtain compound **4a**–**4g**.

### 2.2. Anticancer Activity Assay and Structure–Activity Relationship (SAR) Analyses

Quercetagetin and all synthesized compounds (**2a**–**2t**, **3a**–**3j** and **4a**–**4g**) were tested for their anticancer activity against four leukemia cell lines K562, K562R, U937 and KG-1 using CCK-8 assay and the results are summarized in Table 1. Compound **2a** exhibited the most potent inhibitor function similar to imatinib and with IC_50_ values 0.159, 0.312, 0.271 and 0.276 µmol for K562, K562R, KG-1 and U937, respectively. In addition, compounds **2b**, **2l**, **2m**, **2o**, **2p**, **2r**, **2s**, **2t**, **3a** and **4a** all showed considerable cytotoxic effects on leukemia cell lines.

For U937 cells, quercetagetin has an IC_50_ value 4.62 µM, and methylation of 4, 8, 9 and 13-OH enhances anticancer activity (compound **2a**, IC_50_ = 0.276 µM) but substitution by other low polar groups reduce this activity (compounds **4a**–**4g**). Furthermore, retaining 7-OH is a crucial factor in antiproliferative activity. In fact, intramolecular hydrogen bonds between the 7-OH and 5-carbonyl of flavonoids benefit molecular stability. In addition, methylation or introduction of other moderate polar hydrogen-bond acceptors to 12-OH also increased anticancer activity but substitution with extremely low or high polar groups weakened this activity (compounds **2c**–**2t**). For K562 cells, like U937, methylation of 4, 8, 9 and 13-OH is also a key factor in improving inhibitory activity (compound **2a** IC_50_ = 0.159 µM) and 7 and 12-OH should remain free. K562R is similar to U937, but the substitution of low or high polar groups and methylation of 12-OH all decrease activity. For KG-1 cells, quercetagetin exhibits no significant cytotoxicity. Introduction of high lipophilic groups can improve activity (compounds **2a**, **2b**, **2l**, **2q**, **2r**, **2s**, **2t**, **3a**, **4a** and **4e**, IC_50_ = 0.271 to 8.592 µM). In summary, improving the lipid solubility of quercetagetin is a rational strategy to enhance the anticancer activity. In addition, 7 and 12-OH or substitution of 12-OH by moderate polar hydrogen-bond acceptors play important roles in the anticancer activities of quercetagetin derivatives (Figure 3).

### 2.3. Compound ***2a***-Induced Apoptosis in K562 Cells

To determine whether compound **2a** can induce apoptosis in K562 cells, an experiment of induced apoptosis by annexin V/7-AAD double staining was conducted. After being exposed to compound **2a** at 0.1, 0.5, 1, 2 and 4 µM for 48 h, the apoptotic proportion of K562 cells treated by compound **2a** increased from 6.02% to 32.16% (Figure 4). The results indicated that compound **2a** at a 4 µM concentration can effectively trigger K562 cell apoptosis.

### 2.4. Compound ***2a*** Effectively Arrested the G2 Phase of K562 Cells

Regarding the fact that compound **2a** can effectively induce the apoptosis of K562 cells, we further evaluated the effects of **2a** on the cell cycle of K562 cells. In Figure 5, it can be observed that compound **2a** can arret the cell cycle of K562 in the G2/M phase in a dose-dependent manner. When treated with a 1 µM concentration of **2a**, the cell population of G2/M phase is 60.75%, which is significantly higher than the DMSO group (37.64%), and treatment with 4 µM concentration of **2a** can increase the cell proportion of the G2/M phase to 77.10%.

### 2.5. Compound ***2a*** Suppresses the Expression of Oncoproteins

Over-expression of oncoproteins Bcl-2 and Mcl-1 led to the evasion of cancer cell apoptosis and promoted cancer cell survival. In fact, over-expression of anti-apoptosis proteins Bcl-2 and Mcl-1 served as the hallmarks of leukaemia and myeloma; for this reason, the discovery of new drugs that can inhibit the expression of Bcl-2 and Mcl-1 is important in curing leukemia. Compound **2a** can effectively induce apoptosis and dysregulate the cell cycle of K562 cells, and its effects on the expression of apoptosis-related proteins (Bcl-2, Mcl-1, Erk1/2, p-Erk1/2, STAT3, p-STAT3 Akt and p-Akt) were further investigated by Western blot. The results (Figure 6) showed that **2a** at **a** 2 µM concentration can significantly down-regulate the expression of Bcl-2, Mcl-1 and p-Erk1/2 in K562 cells in a dose-dependent manner. However, the expression of phosphorylated STAT3 and Akt was enhanced, indicating that the anticancer activities of compound **2a** were highlighted by multiple targets. The above results show that **2a** can decrease K562 cell growth in part by lowering the expression levels of Bcl-2, p-Erk1/2, and Mcl-1.

### 2.6. Molecular Docking Analysis

Based on Western blot assay, which showed that compound **2a** exhibited potent Bcl-2 and Mcl-1 inhibitory effects, a molecular docking study was conducted to predict the interaction between **2a** and oncoproteins. The results show (Figure 7) that **2a** presented a high affinity for forming favorable interactions between proteins and the values of bonding energy were −9.375 kcal/mol and −8.324 kcal/mol for Bcl-2 and Mcl-1, respectively. The combination between **2a** and the pocket of Bcl-2 was suitable and the 12-OH of **2a** demonstrated hydrogen bond interaction with Lue267 (2.2 Å), while the B ring of **2a** demonstrated π-π stacking interaction with Phe270. Moreover, the amino acid residues Phe254, Val253, Met250, Val249, Leu246 and Leu267 demonstrated hydrophobic interactions with **2a**.

As depicted in Figure 8, the 12-OH of **2a** demonstrated hydrogen bond interaction with Lue174 (2.2 Å) and the amino acid residues Phe89, Val92, Met116, Val115, Phe112, Phe171, Trp173, Leu174 and Leu176 demonstrated hydrophobic interactions with **2a**. In conclusion, the molecular docking studies provided insights into compound **2a’s** binding capacity and interactions with Bcl-2 and Mcl-1.

### 2.7. ’BOILED-Egg Model’ Analyses and RO5 Evaluation

We performed an analysis on a group of newly synthesized compounds (**1**, **2a**, **2b**, **2l**, **2m**, **2o**, **2p**, **2q**, **2r**, **2s**, **2t**, **3a**, **4a** and **4e**) using the ‘BOILED-egg model’ via the ‘SwissADME’ webserver (http://www.swissadme.ch/index.php (accessed on 24 October 2024)). The current model supplied a clear visual representation that combines speed and accuracy based on the polarity and lipophilicity of the chemical components. As shown in Figure 9, the white area indicates a high likelihood of passive absorption in the gastrointestinal tract, while the yellow region (the yolk) signifies a strong probability of penetrating the blood–brain barrier. Additionally, the blue colour suggests that the compounds were actively effluxed by P-glycoprotein, denoted as (PGP^+^), whereas the red colour implies that P-glycoprotein was not actively effluxing the molecule, symbolized by (PGP^−^). The results (Figure 9) show that the synthesized compounds **2a**, **2b**, **2l**, **2m**, **2o**, **2p**, **2q**, **2r**, **2s**, **2t**, **3a**, **4a** and **4e** are suitable drug candidates and were all distributed in the white area. It is worth noting that compound **1**, the starting material, was taken from the BOILED-egg model, and all compounds exhibited negative P-glycoprotein profiles.

Some important ADME properties of synthesized active compounds were predicted to evaluate the Lipinski rule of 5 (the rules are as follows: mol_MW (molecular weight of the molecule) < 500, QPlogPo/w (predicted octanol/water partition coefficient) < 5, donorHB (hydrogen-bond donor atoms) ≤ 5, accptHB (hydrogen-bond acceptor atoms) ≤ 10 and RotabB (rotatable bonds) < 10) via the ‘SwissADME’ webserver (http://www.swissadme.ch/index.php (accessed on 24 October 2024)) and the results are presented in Table 2. All active compounds except **4e** were considered drug-like and had higher GI absorption compared with compound **1**.

## 3. Discussion

Among the synthesized compounds, a subset demonstrated anticancer activity. Notably, compound **2a** exhibited broad-spectrum and potent anticancer activity against four leukemia cell lines (IC_50_ = 0.276, 0.159, 0.312 and 0.271 µM, respectively), and the SAR analysis suggests that methylation enhanced anticancer activity. These results match with Shi’s study [37]. In addition, the 3,7-dimethyl quercetagetin displayed potent antioxidant activity and demonstrated inhibitory activity against both AChE and BChE [38], reinforcing the critical role of methylation in optimizing bioactivity. Further, the flow cytometry assay indicated that **2a** can trigger apoptosis in K562 cells and arrest the G2/M phase of the cell cycle effectively. In addition, mechanistic experiments showed that **2a** suppresses the expression of Bcl-2, Mcl-1 and p-Erk1/2, indicating that **2a** is a potential inhibitor of these oncoproteins. However, the expression of p-STAT3 and p-Akt was increased and in Sohee’s [39] study, it was reported that quercetagetin can effectively attenuate the expression of p-Akt. Therefore, we can conclude that the methylation of quercetagetin changed the mechanism of anticancer activity, or rather the activity of natural products that is common in a complex mechanism. In the future, we will conduct comprehensive experiments to verify these mechanisms. Molecular docking studies can anticipate the bonding mode between the molecule and protein; our results showed the crucial interactions between **2a** and apoptotic proteins. The methylation of quercetagetin results in hydrophobic interactions with proteins and supports the conclusions of in vitro experiments.

Finally, in silico studies predicted various ADME characteristics of these active derivatives, and **2a** has a higher GI absorption value than quercetagetin and is a suitable drug candidate for future Bcl-2 and Mcl-1 inhibitor studies.

## 4. Materials and Methods

### 4.1. Reactants and Equipment

All the reactants were purchased from Shanghai Aladdin Biochemical Technology Co., Ltd. (Shanghai, China). HRMS data were obtained in the ESI mode on an Agilent 6530 Q-TOF/MS system(Agilent, Santa Clara, CA, USA). ^1^H NMR and ^13^C NMR spectra were recorded on a Bruker 400 MHz spectrometer. Coupling constants (J) are expressed in hertz (Hz). Chemical shifts (δ) of NMR are reported in parts per million (ppm) units relative to an internal control (TSM). ^1^H NMR and ^13^C NMR spectra (S1–S72) of compounds **2**–**4** together with HSQC, HMBC, ^1^H-^1^H COSY spectra of compounds **2b** and **3b**(S73–S78) can be found in Appendix A.

### 4.2. NMR and HRMS Spectral Analysis

**5-hydroxy-2-(3-hydroxy-4-methoxyphenyl)-3**,**6**,**7-trimethoxy-4H-chromen-4-one (2a)**. Yellow green crystal (32.8%). ^1^H NMR (CDCl_3_, 400 MHz) δ: 12.61 (s, 1H-7), 7.73-7.71 (m, 1H-15), 6.97 (d, J = 8.52 Hz, 1H-14), 6.51 (s, 1H-1), 4.00 (s, 3H-16), 3.97 (s, 3H-18), 3.93 (s, 3H-17), 3.88 (s, 3H-19); ^13^C NMR (CDCl_3_, 100 MHz) δ: 178.96-C5, 158.80-C9, 155.66-C3, 152.72-C2, 152.32-C7, 148.87-C12, 145.62-C13, 138.99-C4, 132.29-C8, 123.58-C15, 121.52-C10, 114.40-C11, 110.43-C14, 106.60-C6, 90.37-C1, 60.85-C16, 60.12-C18, 56.30-C17, 56.04-C19; HRMS calcd for C_19_H_19_O_8_ [M+H]^+^ 375.1074 found 375.1082.

**Ethyl-2-(5-(5-hydroxy-3**,**6**,**7-trimethoxy-4-oxo-4H-chromen-2-yl)-2-methoxyphenoxy)acetate (2b)**. Pale green solid (72.6%). ^1^H NMR (CDCl_3_, 400 MHz) δ:12.59 (s, 1H-7), 7.81 (dd, J = 2.00, 8.60 Hz, 2H-15 and 11), 7.70 (d, J = 2.00 Hz, 1H-14), 6.51 (s, 1H-1), 4.78 (s, 2H-16), 4.33-4.28 (m, 2H-22), 3.99 (s, 3H-17), 3.98 (s, 3H-18), 3.93 (s, 3H-20), 3.87 (s, 3H-19), 1.33 (t, J = 7.12 Hz, 3H-23); ^13^C NMR (CDCl_3_, 100 MHz) δ: 178.88-C5, 168.67-C21, 158.82-C9, 152.75-C3, 152.29-C2, 151.98-C7, 146.96-C12, 138.89-C13, 132.30-C-4, 123.53-C8, 122.83-C15, 114.44-C10, 111.60-C11, 106.57-C14, 103.78-C6, 90.33-C1, 66.53-C16, 61.47-C22, 60.90-C17, 60.20-C19, 56.34-C18, 56.05-C20, 14.22-C23; HRMS calcd for C_23_H_25_O_10_ [M+H]^+^ 461.1442 found 461.1437.

We tested the 2D NMR of compound **2b** (Figure 10) to obtain more precise structure information and close inspection of the ^1^H-, ^13^C- and HSQC-NMR spectra displayed five sp^3^ methyl carbons (C-17, C-18, C-19, C-20 and C-23), ten sp^2^ oxygenated quaternary carbons (C-2, C-3, C-4, C-5, C-7, C-8, C-9, C-12, C-13 and C-21), four aromatic carbons (C-1, C-11, C-14 and C-15), two sp^2^ quaternary carbons (C-6 and C-10) and two sp^3^ methylenes (C-16 and C-22). The H-H COSY spectra of **3b** can be divided into two structural units: H15(δ_H_ 7.81)⇔H14 (δ_H_ 7.70) and H22 (δ_H_ 4.33-4.28)⇔H23 (δ_H_ 1.33). The HMBC spectrum showed key correlations of H-7 (δ_H_ 12.57)/C-7 (δ_C_ 152.82), C-6 (δ_C_ 106.70), C-5 (δ_C_ 178.59) and C-8(δ_C_ 132.43); H-16 (δ_H_ 4.76)/C-21 (δ_C_ 168.74), C-12 (δ_C_ 147.10); H-22 (δ_H_ 14.56-14.42)/C-21 (δ_C_ 168.74), C-23 (δ_C_ 14.56); H-17 (δ_H_ 3.96)/C-13 (δ_C_ 148.84); H-18 (δ_H_ 3.85)/C-4 (δ_C_ 138.99); H-19 (δ_H_ 3.91)/C-6 (δ_C_ 132.43); H-20 (δ_H_ 3.96)/C-9 (δ_C_ 158.87).

**(2-(5-(5-hydroxy-3**,**6**,**7-trimethoxy-4-oxo-4H-chromen-2-yl)-2-methoxyphenoxy)acetyl)glycine (2c)**. Light yellow solid (33.8%). ^1^H NMR (CDCl_3_, 400 MHz) δ: 12.57 (s, 1H-7), 7.92 (dd, J = 1.84, 8.64 Hz, 1H-22), 7.69 (d, J = 1.84 Hz, 1H-15), 7.55 (d, J = 8.60 Hz, 1H-14), 7.05 (d, J = 8.76, 1H-11), 6.54 (s, 1H-1), 4.69 (s, 2H-16), 4.19 (d, J = 5.4 Hz, 2H-23), 4.01 (s, 3H-18), 3.94 (s, 3H-19), 3.90 (s, 3H-17), 3.80 (s, 3H-20); ^13^C NMR (CDCl_3_, 100 MHz) δ: 178.86-C5, 169.84-C21, 168.61-C24, 158.89-C9, 152.75-C3, 152.26-C2, 152.04-C4, 146.90-C7, 139.02-C12, 132.35-C13, 124.46-C8, 123.98-C14, 123.18-C15, 119.10-C8, 115.08-C11, 111.66-C10, 106.58-C6, 90.41-C1, 69.36-C16, 60.90-C18, 60.18-C19, 56.43-C20, 56.04-C17, 40.83-C23; HRMS calcd for C_23_H_22_NO_11_ [M+H]^+^ 476.1187 found 476.1192.

**(2-(5-(5-hydroxy-3**,**6**,**7-trimethoxy-4-oxo-4H-chromen-2-yl)-2-methoxyphenoxy)acetyl)alanine (2d)**. Light yellow solid (47.4%). ^1^H NMR (C_2_D_6_SO, 400 MHz) δ: 12.58 (s, 1H-7), 8.35 (d, J = 7.32 Hz, 1H-22), 7.80 (dd, J = 2.04, 8.64 Hz, 1H-15), 7.67 (d, J = 2.08 Hz, 1H-15), 7.21 (d, J = 8.8 Hz, 1H-11), 6.92 (s, 1H-1), 4.65 (s, 2H-16), 4.31 (p, J = 7.28, 1H-23), 3.93 (s, 3H-18), 3.91 (s, 3H-19), 3.82 (s, 3H-17), 3.74 (s, 3H-20), 1.33 (d, J = 7.24 Hz, 3H-24); ^13^C NMR (C_2_D_6_SO, 100 MHz) δ: 178.77-C5, 174.21-C21, 167.78-C25, 159.18-C9, 155.54-C3, 152.26-C2, 152.10-C7, 147.58-C4, 138.61-C12, 132.09-C8, 123.64-C15, 122.57-C14, 114.43-C11, 112.52-C10, 106.13-C6, 91.94-C1, 68.47-C16, 60.53-C18, 60.24-C19, 56.95-C17, 56.29-C20, 47.84-C23, 17.61-C24; HRMS calcd for C_24_H_26_NO_11_ [M+H]^+^ 504.1500 found 504.1489.

**(2-(5-(5-hydroxy-3**,**6**,**7-trimethoxy-4-oxo-4H-chromen-2-yl)-2-methoxyphenoxy)acetyl)-L-valine (2e)**. Light yellow solid (48.2%). ^1^H NMR (CDCl_3_, 400 MHz) δ: 12.53 (s, 1H-7), 7.90-7.87 (m, 1H-22), 7.71 (d, J = 2.4 Hz, 1H-15), 7.59 (d, J = 8.6 Hz, 1H-14), 7.03 (d, J = 5.2 Hz, 1H-11), 6.52 (s, 1H-1), 4.75-4.63 (m, 2H-16 and 23), 3.98 (s, 3H-18), 3.96 (s, 3H-19), 3.93 (s, 3H-18), 3.87 (s, 3H-17), 2.37-2.29 (m, 1H-24), 1.02 (t, J = 8.4 Hz, 6H-27 and 26) (s, 3H-24); ^13^C NMR (CDCl_3_, 100 MHz) δ: 178.84-C5, 174.66-C21, 168.96-C25, 158.91-C9, 154.93-C3, 152.65-C5, 152.26-C4, 147.05-C7, 138.98-C12, 124.53-C15, 123.16-C14, 115.69-C10, 111.65-C11, 106.52-C6, 90.44-C1, 69.77-C16, 60.88-C18, 60.18-C19, 56.79-C17, 54.41-C20, 56.03-C23, 30.94-C24, 19.06-C26, 17.47-C27; HRMS calcd for C_26_H_30_NO_11_ [M+H]^+^ 532.1813 found 532.1806.

**(2-(5-(5-hydroxy-3**,**6**,**7-trimethoxy-4-oxo-4H-chromen-2-yl)-2-methoxyphenoxy)acetyl)leucine (2f)**. Light yellow solid (42.9%). ^1^H NMR (CDCl_3_, 400 MHz) δ: 12.56 (s, 1H-7), 7.91 (dd, J = 2.04, 8.68 Hz, 1H-22), 7.70 (d, J = 2.04 Hz, 1H-15), 7.66 (d, J = 8.16 Hz, 1H-14), 7.06 (d, J = 8.72 Hz, 1H-11), 6.54 (s, 1H-1), 4.81-4.81 (m, 1H-23), 4.72-4.63 (m, 2H-16), 4.01 (s, 3H-18), 3.93 (s, 3H-19), 3.90 (s, 3H-17), 3.78 (s, 3H-20), 2.59-2.51 (m, 2H-24), 2.28-2.20 (m, 1H-26), 2.09 (s, 3H-27), 1.30 (t, J = 12.6 Hz, 3H-28); ^13^C NMR (CDCl_3_, 100 MHz) δ: 178.65-C5, 171.87-C21, 168.31-C25, 158.89-C9, 154.85-C3, 152.74-C2, 152.26-C4, 147.00-C7, 139.03-C12, 132.35-C13, 124.55-C8, 123.25-C14, 115.51-C11, 111.72-C10, 106.58-C6, 90.42-C1, 69.72-C16, 60.88-C18, 60.19-C19, 56.42-C20, 56.07-C17, 52.59-C23, 31.72-C24, 29.84-C26, 15.43 (×2)-C27 and C28; HRMS calcd for C_27_H_32_NO_11_ [M+H]^+^ 546.1970 found 546.1959.

**(2-(5-(5-hydroxy-3**,**6**,**7-trimethoxy-4-oxo-4H-chromen-2-yl)-2-methoxyphenoxy)acetyl)phenylalanine (2g)**. Light yellow solid (39.5%). ^1^H NMR (CDCl_3_, 400 MHz) δ: 12.53 (s, 1H-5), 7.88 (dd, J = 2.00, 8.64 Hz, 1H-15), 7.65 (d, J = 2.00 Hz, 1H-14), 7.57 (d, J = 7.76 Hz, 1H-11), 7.25-7.22 (m, 2H-27 and 31), 7.19-7.17 (m, 2H-28 and 30), 6.99 (d, J = 8.76 Hz, 1H-29), 6.53 (s, 1H-1), 5.01-4.96 (m, 1H-23), 4.64 (s, 2H-16), 3.98 (s, 3H-18), 3.94 (s, 3H-19), 3.86 (s, 3H-17), 3.83 (s, 3H-20), 3.31-3.17 (m, 2H-24); ^13^C NMR (CDCl_3_, 100 MHz) δ: 178.85-C5, 168.85-C21, 156.92-C9, 152.68-C3, 152.26-C2, 152.06-C4, 146.99-C7, 135.59-C12, 129.33-C13, 128.63-C8, 124.46-C14, 123.11-C15, 115.65-C11, 111.67-C10, 106.53-C6, 90.44-C1, 69.70-C16, 60.90-C18, 60.18-C19, 56.42-C17, 55.83-C20, 37.53-C23, 29.71-C24; HRMS calcd for C_30_H_30_NO_11_ [M+H]^+^ 580.1813 found 580.1819.

**(2-(5-(5-hydroxy-3**,**6**,**7-trimethoxy-4-oxo-4H-chromen-2-yl)-2-methoxyphenoxy)acetyl)methionine (2h)**. Light yellow solid (58.2%). ^1^H NMR (CDCl_3_, 400 MHz) δ: 12.48, (s, 1H-7), 7.86 (d, J = 8.84 Hz, 1H-22), 7.71 (s, 1H-15), 7.02 (d, J = 8.80 Hz, 1H-14), 6.82 (d, J = 7.48 Hz, 1H-11), 6.53 (s, 1H-1), 4.83-4.76 (m, 1H-23), 4.68 (s, 2H-16), 3.97 (s, 6H-18 and 19), 3.91 (s, 3H-17), 3.85 (s, 3H-20), 2.56 (t, J = 7.48 Hz, 2H-25), 2.09 (s, 3H-27), 2.08 (s, 2H-26); ^13^C NMR (CDCl_3_, 100 MHz) δ: 178.62-C5, 171.63-C21, 169.13-C24, 158.95-C9, 155.03-C3, 152.55-C2, 146.97-C4, 138.93-C7, 132.29-C12, 124.51-C13, 123.12-C8, 115.82-C14, 111.75-C15, 111.70-C11, 106.47-C6, 90.54-C1, 69.70-C16, 60.68-C18, 60.21-C19, 65.44-C17, 56.04-C20, 41.07-C23, 30.03-C25, 15.38-C27; HRMS calcd for C_26_H_28_NO_9_S [M+H]^+^ 506.1479 found 506.1471.

**(2-(5-(5-hydroxy-3**,**6**,**7-trimethoxy-4-oxo-4H-chromen-2-yl)-2-methoxyphenoxy)acetyl)glutamic acid (2i)**. Light yellow solid (46.5%). ^1^H NMR (C_2_D_6_SO, 400 MHz) δ: 12.57 (s, 1H-7), 8.55 (d, J = 7.8 Hz, 1H-22), 8.19 (d, J = 7.68 Hz, 1H-28), 7.82 (d, J = 1.72 Hz, 1H-15), 7.65 (d, J = 1.8 Hz, 1H-15), 7.20 (d, J = 8.76 Hz, 1H-11), 6.99 (s, 1H-1), 4.69 (d, J = 4.32 Hz, 2H-16), 4.32-4.27 (m, 1H-23), 3.92 (s, 3H-18), 3.90 (s, 3H-19), 3.81 (s, 3H-17), 3.73 (s, 3H-20), 2.27-2.25 (m, 2H-25), 2.05-1.93 (m, 2H-26); ^13^C NMR (C_2_D_6_SO, 100 MHz) δ: 178.78-C5, 174.17-C21, 170.05-C24, 168.33-C27, 159.17-C9, 155.50-C3, 152.28-C2, 152.01-C4, 147.57-C7, 138.61-C12, 132.04-C13, 123.63-C8, 122.56-C14, 114.02-C15, 112.52-C8, 106.11-C11, 92.12-C6, 68.19-C16, 60.54-C18, 60.22-C19, 57.05-C17, 56.29-C20, 51.64-C23, 27.75-C25, 22.76-C26; HRMS calcd for C_26_H_28_NO_13_ [M+H]^+^ 562.1555 found 562.1559.

**6-(2-(5-(5-hydroxy-3**,**6**,**7-trimethoxy-4-oxo-4H-chromen-2-yl)-2-methoxyphenoxy)acetamido)hexanoic acid (2j)**. Light yellow solid (47.1%). ^1^H NMR (CDCl_3_, 400 MHz) δ: 12.56 (s, 1H-7), 7.90 (dd, J = 2.04, 8.68 Hz, 1H-15), 7.67 (d, J = 2.04, 1H-14), 7.05-7.01 (m, 1H-11), 6.54 (s, 1H-1), 4.64 (s, 2H-16), 4.00 (s, 3H-18), 4.00 (s, 3H-19), 3.94 (s, 3H-17), 3.90 (s, 3H-20), 3.41-3.36 (m, 2H-23), 2.36 (t, J = 7.40 Hz, 2H-24), 1.67-1.40 (m, 2H-28), 1.39-1.35 (m, 6H-25, 26 and 27); ^13^C NMR (CDCl_3_, 100 MHz) δ: 178.86-C5, 178.29-C21, 168.27-C29, 158.92-C9, 154.85-C3, 152.73-C2, 151.77-C4, 146.87-C7, 139.04-C12, 132.35-C13, 124.27-C8, 123.30-C10, 114.78-C15, 111.62-C14, 106.56-C6, 90.42-C1, 69.32-C16, 60.90-C18, 60.19-C19, 56.44-C17, 56.01-C20, 38.98-C23, 33.73-C28, 29.27-C24, 28.62-C25, 26.42-C26, 24.53-C27; HRMS calcd for C_27_H_32_NO_11_ [M+H]^+^ 546.1970 found 546.1962.

**2-(3-ethoxy-4-methoxyphenyl)-5-hydroxy-3**,**6**,**7-trimethoxy-4H-chromen-4-one (2k)**. Light yellow solid (72.6%). ^1^H NMR (CDCl_3_, 400 MHz) δ: 12.63 (s, 1H-7), 7.74-7.71 (m, 2H-11 and 15), 7.01 (d, J = 7.52 Hz, 1H-14), 6.51 (s, 1H-1), 4.23-4.18 (m, 2H-16), 3.98 (s, 6H-18 and 19), 3.94 (s, 3H-17), 3.88 (s, 3H-20), 1.54 (t, J = 7.00 Hz, 3H-21); ^13^C NMR (CDCl_3_, 100 MHz) δ: 178.90-C5, 158.77-C9, 155.98-C3, 152.78-C2, 152.33-C7, 151.75-C13, 148.06-C12, 138.80-C8, 132.31-C4, 122.85-C15, 122.06-C14, 112.82-C11, 111.08-C10, 106.60-C6, 90.34-C1, 64.63-C16, 60.88-C17, 60.18-C18, 56.33-C19, 56.02-C20, 14.74-C21; HRMS calcd for C_21_H_23_O_8_ [M+H]^+^ 403.1387 found 403.1379.

**5-hydroxy-3**,**6**,**7-trimethoxy-2-(4-methoxy-3-propoxyphenyl)-4H-chromen-4-one (2l)**. Light yellow solid (68.9%). ^1^H NMR (CDCl_3_, 400 MHz) δ: 12.62 (s, 1H-7), 7.71-7.69 (m, 2H-15 and 11), 6.99 (d, J = 9.16 Hz, 1H-14), 6.50 (s, 1H-1), 4.07 (t, J = 6.76 Hz, 2H-16), 3.97 (s, 3H-18), 3.96 (s, 3H-19), 3.93 (s, 3H-20), 3.86 (s, 3H-17), 1.97-1.88 (m, 2H-21), 1.09 (t, J = 7.40 Hz, 3H-22); ^13^C NMR (CDCl_3_, 100 MHz) δ: 178.87-C5, 158.75-C9, 156.02-C3, 152.74-C2, 152.31-C7, 151.89-C13, 149.28-C12, 138.76-C4, 132.28-C8, 122.81-C15, 122.03-C14, 113.06-C11, 111.19-C10, 106.56-C6, 90.36-C1, 70.75-C16, 60.86-C19, 60.19-C20, 56.33-C17, 56.02-C18, 22.43-C21, 10.46-C22; HRMS calcd for C_22_H_25_O_8_ [M+H]^+^ 417.1544 found 417.1552.

**2-(3-(allyloxy)-4-methoxyphenyl)-5-hydroxy-3**,**6**,**7-trimethoxy-4H-chromen-4-one (2m)**. Light yellow solid (70.6%), ^1^H NMR (CDCl_3_, 400 MHz) δ: 12.61 (s, 1H-7), 7.74-7.71 (m, 2H-15 and 11), 7.00 (d, J = 9.12 Hz, 1H-14), 6.49 (s, 1H-1), 6.17-6.10 (m, 1H-21), 5.48 (dd, J = 1.44, 17.28 Hz, 1H-22), 5.35 (dd, J = 1.28, 10.48 Hz, 1H-22), 4.70 (d, J = 5.44 Hz, 2H-16), 3.97 (s, 3H-18), 3.97 (s, 3H-19), 3.93 (s, 3H-20), 3.86 (s, 3H-17); ^13^C NMR (CDCl_3_, 100 MHz) δ: 178.87-C5, 158.77-C9, 155.80-C3, 152.74-C2, 152.29-C7, 151.88-C12, 147.68-C13, 138.80-C4, 132.94-C8, 132.29-C21, 122.78-C22, 122.35-C15, 118.39-C14, 113.58-C11, 111.18-C10, 106.58-C6, 90.33-C1, 70.07-C16, 60.86-C18, 60.21-C19, 56.32-C17, 56.00-C20; HRMS calcd for (C_22_H_22_O_8_+H)^+^ 415.1387 found 415.1391.

**2-(3-butoxy-4-methoxyphenyl)-5-hydroxy-3**,**6**,**7-trimethoxy-4H-chromen-4-one (2n)**. Light yellow solid (71.2%). ^1^H NMR (CDCl_3_, 400 MHz) δ: 12.63 (s, 1H-7), 7.73-7.71 (m, 2H-11 and 15), 7.00 (d, J = 9.16 Hz, 1H-14), 6.52 (s, 1H-1), 4.13 (t, J = 6.72 Hz, 2H-16), 3.98 (d, J = 4.00, 6H-18 and 19), 3.94 (s, 3H-20), 3.88 (s, 3H-17), 1.93-1.86 (m, 2H-21), 1.60-1.51 (m, 2H-22), 1.03 (t, J = 7.36 Hz, 3H-23); ^13^C NMR (CDCl_3_, 100 MHz) δ: 178.90-C5, 158.76-C9, 156.04-C3, 152.78-C2, 152.34-C7, 151.89-C12, 148.33-C13, 138.79-C8, 132.31-C4, 122.64-C15, 122.00-C14, 113.04-C11, 111.19-C10, 106.61-C6, 90.35-C1, 69.00-C16, 60.88-C18, 60.19-C19, 56.33-C17, 56.04-C20, 31.16-C21, 19.23-C22, 13.89-C23; HRMS calcd for C_23_H_27_O_8_ [M+H]^+^ 431.1700 found 431.1694.

**5-hydroxy-3**,**6**,**7-trimethoxy-2-(4-methoxy-3-(pentyloxy)phenyl)-4H-chromen-4-one (2o)**. Light yellow solid (67.1%). ^1^H NMR (CDCl_3_, 400 MHz) δ: 12.63 (s, 1H-7), 7.73-7.70 (m, 2H-11 and 15), 7.00 (d, J = 9.16 Hz, 1H-14), 6.52 (s, 1H-1), 4.11 (t, J = 6.88 Hz, 2H-16), 3.98 (s, 3H-18), 3.97 (s, 3H-19), 3.94 (s, 3H-20), 3.87 (s, 3H-17), 1.95-1.88 (m, 2H-21), 1.52-1.40 (m, 4H-22 and 23), 0.96 (t, J = 7.16 Hz, 3H-24); ^13^C NMR (CDCl_3_, 100 MHz) δ: 178.89-C5, 158.76-C9, 156.04-C3, 152.78-C2, 152.33-C7, 151.88-C12, 148.31-C13, 138.79-C4, 132.30-C8, 122.83-C15, 122.01-C14, 113.03-C11, 111.18-C10, 106.60-C6, 90.35-C1, 69.31-C16, 60.88-C18, 60.19-C19, 56.33-C20, 56.03-C17, 28.82-C21, 28.12-C22, 22.50-C23, 14.02-C24; HRMS calcd for C_24_H_29_O_8_ [M+H]^+^ 445.1857 found 445.1861.

**5-hydroxy-3**,**6**,**7-trimethoxy-2-(4-methoxy-3-(pent-4-en-1-yloxy)phenyl)-4H-chromen-4-one (2p)**. Yellow green solid (75.8%). ^1^H NMR (CDCl_3_, 400 MHz) δ: 12.63 (s, 1H-7), 7.73-7.71 (m, 2H-15 and 11), 7.00 (d, J = 8.32 Hz, 1H-14), 6.51 (s, 1H-1), 5.95-5.84 (m, 1H-23), 5.13-5.02 (m, 2H-24), 4.13 (t, J = 6.68 Hz, 2H-16), 3.98 (s, 3H-18), 3.97 (s, 3H-19), 3.94 (s, 3H-20), 3.87 (s, 3H-17), 2.33-2.28 (m, 2H-21), 2.01 (p, J = 6.72 Hz, 2H-22); ^13^C NMR (CDCl_3_, 100 MHz) δ: 178.89-C5, 158.77-C9, 155.97-C3, 152.77-C2, 152.32-C7, 151.93-C12, 148.23-C13, 138.79-C4, 137.69-C8, 132.30-C23, 122.84-C24, 122.15-C14, 115.29-C15, 113.24-C11, 111.23-C10, 106.59-C6, 90.35-C1, 68.59-C16, 60.88-C18, 60.18-C19, 56.33-C20, 56.03-C17, 30.05-C21, 28.22-C22; HRMS calcd for C_24_H_27_O_8_ [M+H]^+^ 443.1700 found 443.1705.

**5-hydroxy-3**,**6**,**7-trimethoxy-2-(4-methoxy-3-((3-methylbut-2-en-1-yl)oxy)phenyl)-4H-chromen-4-one (2q)**. Yellow green solid (66.5%). ^1^H NMR (CDCl_3_, 400 MHz) δ: 12.63 (s, 1H-7), 7.75-7.70 (m, 2H-15 and 11), 7.00 (d, J = 8.28 Hz, 1H-14), 6.51 (s, 1H-1), 5.59-5.56 (m, 1H-21), 4.69 (d, J = 6.64 Hz, 2H-16), 3.98 (d, J = 1.6 Hz, 6H-18 and 19), 3.94 (s, 3H-20), 3.87 (s, 3H-17), 1.61 (d, J = 7.08 Hz, 6H-23 and 24); ^13^C NMR (CDCl_3_, 100 MHz) δ: 178.90-C5, 158.76-C9, 156.01-C3, 152.78-C2, 152.33-C7, 151.92-C12, 148.01-C13, 138.81-C4, 138.19-C8, 132.30-C21, 122.77-C22, 122.14-C15, 119.62-C14, 113.22-C11, 111.03-C10, 106.60-C6, 90.31-C1, 66.03-C16, 60.88-C18, 60.17-C19, 56.32-C20, 55.99-C17, 25.89-C23, 18.31-C24; HRMS calcd for C_24_H_27_O_8_ [M+H]^+^ 443.1700 found 443.1693.

**5-hydroxy-2-(3-isopropoxy-4-methoxyphenyl)-3**,**6**,**7-trimethoxy-4H-chromen-4-one (2r)**. Light yellow solid (63.2%). ^1^H NMR (CDCl_3_, 400 MHz) δ: 12.63 (s, 1H-7), 7.75-7.70 (m, 2H-15 and 11), 7.01 (d, J = 8.60 Hz, 1H-14), 6.52 (s, 1H-1), 4.73-4.60 (m, 1H-16), 3.98 (s, 3H-18), 3.96 (s, 3H-19), 3.94 (s, 3H-20), 3.88 (s, 3H-17), 1.45 (t, J = 4.84 Hz, 6H-21 and 22); ^13^C NMR (CDCl_3_, 100 MHz) δ: 178.90-C5, 158.77-C9, 156.00-C3, 152.85-C2, 152.78-C7, 152.34-C12, 146.98-C13, 138.77-C4, 132.30-C8, 122.61-C15, 122.30-C14, 116.07-C11, 111.48-C10, 106.61-C6, 90.35-C1, 71.83-C16, 60.89-C19, 60.19-C20, 56.33-C18, 56.00-C17, 22.03 (×2)-C21 and C22; HRMS calcd for C_22_H_25_O_8_ [M+H]^+^ 417.1544 found 417.1548.

**5-hydroxy-2-(3-isobutoxy-4-methoxyphenyl)-3**,**6**,**7-trimethoxy-4H-chromen-4-one (2s)**. Light yellow solid (69.6%). ^1^H NMR (CDCl_3_, 400 MHz) δ: 12.63 (s, 1H-7), 7.72-7.69 (m, 2H-15 and 11), 7.00 (d, J = 9.04 Hz, 1H-14), 6.52(s, 1H-1), 3.98 (s, 6H-18 and 19), 3.96 (s, 3H-20), 3.94-3.86 (m, 5H-17 and 16), 2.27-2.17 (m, 1H-21), 1.09 (d, J = 6.68 Hz, 6H-22 and 23); ^13^C NMR (CDCl_3_, 100 MHz) δ: 178.69-C5, 156.76-C9, 156.09-C3, 152.76-C2, 152.34-C7, 152.04-C12, 148.52-C13, 138.76-C4, 132.29-C8, 122.60-C15, 122.03-C14, 113.35-C11, 111.35-C10, 106.59-C6, 90.38-C1, 75.79-C16, 60.68-C18, 60.19-C19, 56.34-C20, 56.07-C17, 26.17-C21, 19.32 (x2)-C22 and C23; HRMS calcd for C_23_H_27_O_8_ [M+H]^+^ 431.1700 found 431.1707.

**Ethyl(E)-4-(5-(5-hydroxy-3**,**6**,**7-trimethoxy-4-oxo-4H-chromen-2-yl)-2-methoxyphenoxy)but-2-enoate (2t)**. Pale green solid (71.8%). ^1^H NMR (CDCl_3_, 400 MHz) δ:12.59 (s, 1H-7), 7.77 (dd, J = 2.00, 8.60 Hz, 2H-15 and 11), 7.68 (d, J = 1.92 Hz, 1H-14), 7.14 (dt, J = 4.28, 15.76 Hz, 1H-21), 6.51 (s, 1H-1), 6.25 (d, J = 15.8 Hz, 1H-22), 4.87-4.86 (m, 2H-16), 4.25-4.20 (m, 2H-24), 3.98 (d, J = 4.24 Hz, 6H-18 and 19), 3.93 (s, 3H-20), 3.85 (s, 3H-17), 1.30 (t, 7.12 Hz, 3H-25); ^13^C NMR (CDCl_3_, 100 MHz) δ: 178.85-C5, 165.89-C23, 158.83-C9, 155.51-C3, 152.29-C2, 152.29-C7, 151.95-C12, 147.22-C13, 142.02-C4, 138.85-C8, 132.34-C21, 123.00-C15, 122.09-C22, 122.68-C14, 113.91-C11, 111.48-C10, 106.58-C6, 90.37-C1, 67.90-C16, 60.86-C18, 60.61-C19, 60.18-C20, 56.35-C17, 56.04-C24, 14.21-C25; HRMS calcd for C_25_H_27_O_10_ [M+H]^+^ 487.1599 found 487.1595.

**2-(3**,**4-dimethoxyphenyl)-5-hydroxy-3**,**6**,**7-trimethoxy-4H-chromen-4-one (3a)**. Yellow green crystal (36.4%). ^1^H NMR (CDCl_3_, 400 MHz) δ: 12.59 (s, 1H-7), 7.70 (d, J = 8.0 Hz, 1H-15), 7.66 (s, 1H-11), 6.97 (d, J = 8.0 Hz, 1H-14), 6.48 (s, 1H-1), 3.95 (s, 9H-16, 17 and 18), 3.90 (s, 3H-19), 3.85 (s, 3H-20); ^13^C NMR (CDCl_3_, 100 MHz) δ: 178.83-C5, 158.76-C9, 155.83-C3, 152.69-C2, 152.26-C7, 151.42-C12, 148.78-C13, 138.77-C4, 132.77-C8, 122.86-C15, 122.15-C10, 111.30-C11, 110.88-C14, 106.52-C6, 90.36-C1, 60.81-C16, 60.12-C17, 56.31-C19, 56.07-C18, 55.97-C20; HRMS calcd for C_20_H_20_O_8_ [M+H]^+^ 389.1231 found 389.1226.

**Ethyl-2-((2-(3**,**4-dimethoxyphenyl)-3**,**6**,**7-trimethoxy-4-oxo-4H-chromen-5-yl)oxy)acetate (3b)**. White green solid (93.1%). ^1^H NMR (CDCl_3_, 400 MHz) δ: 7.68 (d, J = 4.0 Hz, 1H-15), 7.66 (s, 1H-11), 6.96 (d, J = 8.0 Hz, 1H-14), 6.75 (s, 1H-1), 4.72 (s, 2H-21), 4.28 (q, J = 7.2 Hz, 2H-23), 3.95 (s, 6H-16 and 17), 3.94 (s, 3H-19), 3.90 (s, 3H-18), 3.81 (s, 3H-20), 3.14 (t, 3H-24); ^13^C NMR (CDCl_3_, 100 MHz) δ: 173.38-C5, 168.92-C22, 157.57-C9, 153.33-C3, 150.91-C2, 149.96-C7, 148.63-C12, 140.73-C14, 139.89-C8, 123.13-C15, 121.72-C10, 112.90-C11, 111.13-C14, 110.75-C14, 96.59-C6, 70.75-C21, 61.79-C23, 60.98-C16, 59.88-C17, 56.36-C19, 56.03-C18, 55.95-C20, 14.21-C24; HRMS calcd for C_24_H_27_O_10_ [M+H]^+^ 475.1599 found 475.1585.

We tested the 2D NMR of **3b** (Figure 11) to gain more precise structural information and close inspection of ^1^H, ^13^C and HMQC-NMR spectra displayed six sp^3^ methyl carbons (C-16, C-17, C-18, C-19, C-20 and C-24), ten sp^2^ oxygenated quaternary carbons (C-2, C-3, C-4, C-5, C-7, C-8, C-9, C-12, C-13 and C-22), four aromatic carbons (C-1, C-11, C-14 and C-15), two sp^2^ quaternary carbons (C-6 and C-10) and two sp^3^ methylenes (C-21 and C23). The H-H COSY spectra of **3b** can be divided into two structural units: H24 (δ_H_ 4.28)⇔H23 (δ_H_ 1.28); H15 (δ_H_ 7.68)⇔H14 (δ_H_ 6.96). The HMBC spectrum showed key correlations of H-16 (δ_H_ 3.95)/C-12 (δ_C_ 149.45), H-17 (δ_H_ 3.94)/C-13 (δ_C_ 149.45), H-18 (δ_H_ 3.90)/C-4 (δ_C_ 140.82), H-21 (δ_H_ 4.72)/C-22 (δ_C_ 168.93) and C-7 (δ_C_ 150.09), H-19 (δ_H_ 3.94)/C-8 (δ_C_ 140.01), H-20 (δ_H_ 3.81)/C-9 (δ_C_ 157.64).

**(2-((2-(3**,**4-dimethoxyphenyl)-3**,**6**,**7-trimethoxy-4-oxo-4H-chromen-5-yl)oxy)acetyl)glycine (3c)**. Light yellow solid (35.4%). ^1^H NMR (C_2_D_6_SO, 400 MHz) δ: 9.64 (t, J = 5.96 Hz, 1H-25), 7.72 (dd, J = 2.04, 8.56 Hz, 1H-15), 7.66 (d, J = 2.0 Hz, 1H-14), 7.22 (s, 1H-11), 7.15 (d, J = 8.72 Hz, 1H-1), 4.87 (s, 2H-21), 3.97 (s, 3H-18), 3.86 (s, 3H-19), 3.80 (s, 3H-16), 3.77 (s, 3H-17), 3.73 (s, 3H-20); ^13^C NMR (C_2_D_6_SO, 100 MHz) δ: 173.58-C5, 171.40-C22, 169.28-C24, 158.36-C9, 153.45-C3, 149.93-C2, 148.93-C7, 140.43-C4, 139.52-C8, 122.67-C12, 122.22-C13, 112.02-C11, 111.63-C14, 111.17-C6, 97.43-C1, 73.43-C21, 61.37-C18, 59.85-C19, 57.18-C16, 56.15-C17, 56.12-C20, 40.84-C23; HRMS calcd for C_24_H_26_NO_11_ [M+H]^+^ 504.1500 found 504.1489.

**(2-((2-(3**,**4-dimethoxyphenyl)-3**,**6**,**7-trimethoxy-4-oxo-4H-chromen-5-yl)oxy)acetyl)alanine (3d)**. Light yellow solid (45.9%). ^1^H NMR (C_2_D_6_SO, 400 MHz) δ: 9.79 (d, J = 7.16 Hz, 1H-26), 7.73 (dd, J = 2.04 Hz, 8.56 Hz, 1H-15), 7.67 (d, J = 2.04 Hz, 1H-14), 7.22 (s, 1H-1), 7.16 (d, J = 8.72 Hz, 1H-11), 4.73-4.64 (m, 2H-21), 4.38-4.31 (m, 1H-23), 3.98 (s, 3H-16), 3.87 (s, 3H-17), 3.87 (s, 3H-18), 3.81 (s, 3H-19), 3.80 (s, 3H-20), 1.49 (d, J = 7.32 Hz, 3H-24); ^13^C NMR (C_2_D_6_SO, 100 MHz) δ:174.34-C5, 173.64-C22, 168.57-C25, 158.42-C9, 153.47-C3, 151.52-C2, 150.06-C7, 148.93-C12, 140.38-C13, 138.26-C4, 122.70-C8, 122.23-C15, 112.03-C10, 111.68-C11, 110.92-C14, 97.23-C1, 73.48-C21, 61.31-C16, 59.81-C17, 56.15-C18, 56.12 (×2)-C19 and C20, 47.74-C23, 17.59-C24; HRMS calcd for C_25_H_28_NO_11_ [M+H]^+^ 518.1657 found 518.1646.

**(2-((2-(3**,**4-dimethoxyphenyl)-3**,**6**,**7-trimethoxy-4-oxo-4H-chromen-5-yl)oxy)acetyl)-L-valine (3e)**. Light yellow solid (34.1%). ^1^H NMR (CDCl_3_, 400 MHz) δ: 7.90 (dd, J = 1.92, 8.68 Hz, 1H-26), 7.68 (d, J = 1.96 Hz, 1H-15), 7.54 (d, J = 8.92 Hz, 1H-14), 7.04 (d, J = 8.68 Hz, 1H-11), 6.52 (s, 1H-1), 4.72-4.64 (m, 2H-21), 4.64-4.61 (m, 1H-23), 4.00 (s, 3H-18), 3.98 (s, 3H-19), 3.92 (s, 3H-16), 3.88 (s, 3H-17), 3.75 (s, 3H-20), 2.26-2.20 (m, 1H-24), 0.96-0.94 (m, 6H-28 and 27); ^13^C NMR (CDCl_3_, 100 MHz) δ:178.84-C5, 171.90-C25, 168.34-C22, 158.87-C9, 154.92-C3, 152.70-C2, 152.24-C7, 152.09-C12, 147.08-C13, 138.99-C4, 132.31-C10, 124.49-C15, 123.17-C14, 115.37-C11, 111.65-C6, 90.41-C1, 69.71-C21, 60.87-C18, 60.18-C19, 56.73-C20, 56.73-C16, 56.41-C17, 56.05-C23, 31.24-C24, 18.95-C27, 17.64-C28; HRMS calcd for C_28_H_34_NO_11_ [M+H]^+^ 546.1970 found 546.1964.

**(2-((2-(3**,**4-dimethoxyphenyl)-3**,**6**,**7-trimethoxy-4-oxo-4H-chromen-5-yl)oxy)acetyl)leucine (3f)**. white yellow solid (52.8%). ^1^H NMR (CDCl_3_, 400 MHz) δ: 10.10 (d, J = 3.28 Hz, 1H-23), 7.71-7.69 (m, 2H-15 and 14), 6.99 (d, J = 9.16 Hz, 1H-11), 6.74 (s, 1H-1), 4.97-4.78 (m, 2H-21), 4.64-4.59 (m, 1H-24), 3.99 (s, 3H-16), 3.98 (s, 3H-17), 3.97 (s, 3H-18), 3.85 (s, 3H-19), 3.84 (s, 3H-20), 1.96-1.84 (m, 2H-26), 1.35 (s, 1H-27), 1.04 (d, J = 6.28 Hz, 3H-28), 0.99 (d, J = 6.2 Hz, 3H-29); ^13^C NMR (CDCl_3_, 100 MHz) δ: 175.65-C5, 174.15-C22, 170.73-C25, 158.15-C9, 153.85-C3, 153.64-C2, 151.18-C7, 150.51-C12, 148.73-C13, 140.61-C4, 137.92-C8, 122.92-C12, 121.86-C10, 111.30-C11, 111.15-C14, 110.84-C6, 95.64-C1, 73.34-C21, 61.20-C17, 59.85-C16, 56.42-C18, 56.08-C19, 55.98-C20, 39.74-C26, 24.85-C27, 23.05-C28, 21.51-C29;HRMS calcd for C_28_H_34_NO_11_ [M+H]^+^ 560.2126 found 560.2122.

**(2-((2-(3**,**4-dimethoxyphenyl)-3**,**6**,**7-trimethoxy-4-oxo-4H-chromen-5-yl)oxy)acetyl)phenylalanine (3g)**. White yellow solid (49.4%). ^1^H NMR (CDCl_3_, 400 MHz) δ: 10.31 (d, J = 7.6 Hz, 1H-26), 7.74-7.71 (m, 1H-), 7.37 (d, J = 7.16, 2H-28 and 32), 7.23 (t, J = 7.6, 2H-29 and 31), 7.17 (d, J = 7.24, 1H-15), 7.01 (d, J = 8.36, 1H-14), 6.75 (s, 1H-1), 4.88-4.73 (m, 3H-21 and 23), 4.00 (s, 3H-18), 3.99 (s, 3H-19), 3.98 (s, 3H-16), 3.84 (s, 3H-17), 3.81 (s, 3H-20), 3.48-3.33 (m, 2H-24); ^13^C NMR (CDCl_3_, 100 MHz) δ: 174.17-C5, 174.02-C22, 170.82-C25,158.17-C9, 153.55-C3, 151.25-C2, 151.25-C7, 150.48-C12, 148.78-C13, 140.65-C4, 138.00-C8, 137.30-C12, 129.35-C30, 128.33-C31, 126.58-C29, 121.91 (x2)-C32 and 28, 111.28-C10, 111.16-C11, 110.87-C6, 95.69-C1, 73.24-C21, 61.19-C24, 60.02-C19, 56.43-C18, 56.11-C17, 56.00-C16, 54.68-C20, 36.86-C23; HRMS calcd for C_31_H_34_NO_11_ [M+H]^+^ 596.2126 found 596.2137.

**(2-((2-(3**,**4-dimethoxyphenyl)-3**,**6**,**7-trimethoxy-4-oxo-4H-chromen-5-yl)oxy)acetyl)methionine (3h)**. Yellow solid (62.6%). ^1^H NMR (CDCl_3_, 400 MHz) δ: 10.34 (d, J = 6,48 Hz, 1H-26), 7.72 (d, J = 7.36 Hz, 2H-15 and 14), 7.00 (d, J = 9.04 Hz, 1H-11), 6.76 (s, 1H-1), 4.95-4.82 (m, 2H-21), 4.69-4.62 (m, 1H-23), 4.01 (s, 3H-18), 3.99 (s, 3H-19), 3.98 (s, 3H-16), 3.86 (s, 3H-17), 3.85 (s, 3H-20), 1.71 (d, J = 7.24, 2H-24), 1.33 (d, J = 19.6 Hz, 2H-27), 1.27 (s, 3H-28); ^13^C NMR (CDCl_3_, 100 MHz) δ: 174.78-C5, 174.19-C22, 170.95-C25, 156.20-C9, 154.02-C3, 153.58-C2, 151.25-C7, 150.44-C12, 148.76-C13, 140.64-C4, 138.01-C8, 121.91-C10, 111.32-C11, 111.12-C14, 110.86-C6, 95.74-C1, 73.25-C21, 61.24-C18, 59.96-C19, 56.46-C16, 56.11-C17, 56.00-C20, 48.69-C23, 29.69-C24, 29.23-C27, 16.58-C28; HRMS calcd for C_27_H_32_NO_11_S [M+H]^+^ 578.1691 found578.1687.

**(2-((2-(3**,**4-dimethoxyphenyl)-3**,**6**,**7-trimethoxy-4-oxo-4H-chromen-5-yl)oxy)acetyl)glutamic acid (3i)**. Light yellow solid (56.3%). ^1^H NMR (C_2_D_6_SO, 400 MHz) δ: 9.76 (d, J = 7.64 Hz, 1H-25), 7.73 (dd, J = 2.08, 8.56 Hz, 1H-15), 7.68 (d, J = 2.04 Hz, 1H-14), 7.23 (s, 1H-11), 7.16 (d, J = 8.76 Hz, 1H-1), 4.75-4.67 (m, 2H-21), 4.40-4.35 (m, 1H-23), 3.98 (s, 3H-18), 3.87 (s, 3H-19), 3.87 (s, 3H-16), 3.80 (s, 3H-17), 3.80 (s, 3H-20), 2.51-2.40 (m, 2H-26), 2.20-2.10 (m, 2H-27); ^13^C NMR (C_2_D_6_SO, 100 MHz) δ: 174.16-C5, 173.77-C22, 173.49-C24, 169.00-C24, 159.45-C9, 153.52-C3, 151.52-C2, 148.94-C7, 140.38-C4, 138.28-C8, 122.71-C12, 122.24-C13, 112.04-C11, 111.70-C14, 110.91-C6, 97.23-C1, 73.39-C21, 61.33-C18, 59.65-C19, 57.20-C23, 56.16-C16, 56.13-C17, 51.47-C20, 30.53-C26, 26.64-C27; HRMS calcd for C_26_H_28_NO_13_ [M+H]^+^ 562.1555 found 562.1542.

**7-(2-((2-(3**,**4-dimethoxyphenyl)-3**,**6**,**7-trimethoxy-4-oxo-4H-chromen-5-yl)oxy)acetamido)heptanoic acid (3j)**. Light yellow solid (48.9%). ^1^H NMR (C_2_D_6_SO, 400 MHz) δ: 9.44 (t, J = 5.48, 1H-21), 7.73 (dd, J = 1.96, 8.52 Hz, 1H-15), 7.67 (d, J = 1.96 Hz, 1H-14), 7.21 (s, 1H-11), 7.16 (d, J = 8.68 Hz, 1H-1), 4.64 (s, 2H-21), 3.98 (s, 3H-18), 3.87 (s, 3H-19), 3.81 (s, 3H-16), 3.78 (s, 3H-17), 3.26-3.22 (m, 2H-23), 2.51 (t, J = 1.68 Hz, 2H-24), 2.20 (t, J = 7.36 Hz, 2H-28), 1.59-1.49 (m, 4H-27 and 25), 1.38-1.34 (m, 2H-26); ^13^C NMR (C_2_D_6_SO, 100 MHz) δ: 174.94-C5, 173.73-C30, 168.45-C29, 158.41-C9, 153.49-C3, 151.52-C2, 148.94-C7, 140.39-C4, 138.30-C8, 122.70-C12, 122.20-C13, 112.04-C11, 111.68-C14, 97.19-C1, 73.57-C21, 69.16-C18, 61.28-C19, 59.82-C20, 57.17-C16, 57.13-C17, 56.16-C23, 56.13-C28, 34.11-C27, 29.31-C26, 28.79-C25, 24.91-C24; HRMS calcd for C_28_H_34_NO_11_ [M+H]^+^ 560.2126 found 560.2131.5

**3**,**6**,**7-triethoxy-2-(4-ethoxy-3-hydroxyphenyl)-5-hydroxy-4H-chromen-4-one (4a)**. Yellow solid (33,2%). ^1^H NMR (CDCl_3_, 400 MHz) δ: 12.60 (s, 1H-7), 7.74-7.70 (m, 2H-15 and 11), 6.93 (d, J = 8.4 Hz, 1H-14), 6.47 (s, 1H-1), 4.22-4.06 (m, 8H-16, 18, 20 and 22), 1.53-1.49 (m, 6H-19 and 21), 1.40 (t, J = 7.04 Hz, 3H-17), 1.35 (t, J = 7.08, 3H-23); ^13^C NMR (CDCl_3_, 100 MHz) δ: 179.12-C5, 156.46-C9, 155.87-C3, 153.03-C2, 152.27-C4, 148.04-C7, 145.52-C12, 137.87-C13, 131.17-C8, 123.64-C10, 121.56-C15, 114.40-C14, 110.97-C11, 106.38-C6, 90.92-C1, 68.87-C18, 68.53-C20, 64.72-C16, 64.64-C22, 15.54-C17, 15.50-C19, 14.75-C21, 14.56-C23; HRMS calcd for C_23_H_27_O_8_ [M+H]^+^ 431.1700 found 431.1692.

**5-hydroxy-2-(3-hydroxy-4-propoxyphenyl)-3**,**6**,**7-tripropoxy-4H-chromen-4-one (4b)**. Yellow oil (35.1%). ^1^H NMR (CDCl_3_, 400 MHz) δ: 12.59 (s, 1H-7), 7.73-7.70 (m, 1H-14), 6.94 (d, J = 8.48 Hz, 2H-15 and 11), 6.49 (s, 1H-1), 4.11 (t, J = 6.56 Hz, 2H-16), 4.05-4.00 (m, 4H-19 and 22), 3.97 (t, J = 6.88 Hz, 2H-25), 1.95-1.89 (m, 4H-17 and 20), 1.83-1.77 (m, 4H-23 and 26), 1.13-1.07 (m, 9H-21, 24 and 18), 0.97 (t, J = 7.40 Hz, 3H-27); ^13^C NMR (CDCl_3_, 100 MHz) δ: 179.10-C5, 158.58-C9, 155.73-C3, 153.00-C2, 152.24-C4, 148.07-C7, 145.51-C12, 138.10-C13, 131.51-C8, 123.69-C15, 121.67-C14, 114.44-C10, 110.98-C11, 106.41-C6, 90.90-C1, 75.03-C19, 74.49-C16, 70.60-C22, 70.53-C25, 23.39-C20, 23.33-C23, 22.46-C17, 22.38-C26, 10.50-C21, 10.44 (×2)-C24 and C18, 10.38-C27; HRMS calcd for C_27_H_35_O_8_ [M+H]^+^ 487.2326 found 487.2321.

**3**,**6**,**7-tris(allyloxy)-2-(4-(allyloxy)-3-hydroxyphenyl)-5-hydroxy-4H-chromen-4-one (4c)**. Yellow oil (32.4%). ^1^H NMR (CDCl_3_, 400 MHz) δ: 12.61 (s, 1H-7), 7.71-7.68 (m, 2H-11 and 15), 6.93 (d, J = 9.32 Hz, 1H-14), 6.47 (s, 1H-1), 6.20-6.03 (m, 3H-17, 20 and 23), 5.98-5.93 (m, 1H-26), 5.50-5.18 (m, 8H-18, 21, 24 and 27), 4.40-4.58 (m, 8H-16, 19, 22 and 25); ^13^C NMR (CDCl_3_, 100 MHz) δ: 178.98-C5, 157.93-C9, 156.01-C3, 153.08-C2, 152.19-C7, 147.80-C12, 145.63-C13, 134.32-C4, 133.40 (x2)-C26 and C23, 132.23 (×2)-C17 and C20, 132.08-C10, 118.92-C24, 118.66-C27, 118.21-C20, 118.10-C17, 114.74-C11, 111.55-C6, 91.47-C1, 74.00-C25, 73.33-C16, 69.82-C22, 69.68-C19; HRMS calcd for C_27_H_27_O_8_ [M+H]^+^ 479.1700 found 479.1689.

**3**,**6**,**7-tributoxy-2-(4-butoxy-3-hydroxyphenyl)-5-hydroxy-4H-chromen-4-one (4d)**. Yellow oil (36.1%). ^1^H NMR (CDCl_3_, 400 MHz) δ: 12.59 (s, 1H-7), 7.74-7.69 (m, 2H-11 and 15), 6.95 (d, J = 6.24 Hz, 1H-14), 6.48 (s, 1H-1), 4.13 (t, J = 6.52 Hz, 2H-20), 4.07-3.98 (m, 6H-16, 24 and 28), 1.92-1.84 (m, 4H-17 and 21), 1.80-1.72 (m, 4H-25 and 29), 1.53-1.49 (m, 6H-18, 21 and 26), 1.34-1.30 (m, 2H-30), 0.96-0.94 (m, 9H-23, 27 and 31), 0.89 (t, J = 7.08 Hz, 3H-19); ^13^C NMR (CDCl_3_, 100 MHz) δ: 179.07-C5, 158.58-C9, 155.76-C3, 152.98-C2, 152.23-C7, 148.12-C12, 145.53-C13, 138.10-C4, 131.51-C8, 123.62-C15, 121.61-C11, 114.44-C14, 110.93-C6, 106.38-C6, 90.87-C1, 73.42-C16, 72.95-C20, 69.07-C24, 29.86-C28, 29.75-C17, 28.80-C21, 28.68-C25, 28.17-C29, 19.20-C18, 19.15-C22, 19.10 (×2)-C26 and C30, 13.91-C31, 13.82-C27, 13.76 (×2)-C19 and C23; HRMS calcd for C_31_H_43_O_8_ [M+H]^+^ 543.2952 found 543.2958.

**5-hydroxy-2-(3-hydroxy-4-(pentyloxy)phenyl)-3**,**6**,**7-tris(pentyloxy)-4H-chromen-4-one (4e)**. Yellow oil (31.7%). ^1^H NMR (CDCl_3_, 400 MHz) δ: 12.59 (s, 1H-7), 7.71-7.70 (m, 1H-15), 7.68 (s, 1H-11), 6.93 (d, J = 8.28 Hz, 1H-14), 6.46 (s, 1H-1), 4.13 (t, J = 6.52 Hz, 2H-16), 4.07-3.98 (m, 6H-21, 26 and 31), 1.91-1.86 (m, 4H-17 and 22), 1.82-1.72 (m, 4H-27 and 32), 1.53-1.39 (m, 12H-18, 19, 23, 24, 28 and 29), 1.32-1.27 (m, 4H-33 and 34), 0.96 (t, J = 7.28 Hz, 9H-20, 25 and 30), 0.89 (t, J = 7.08 Hz, 3H-35); ^13^C NMR (CDCl_3_, 100 MHz) δ: 179.07-C5, 158.58-C9, 155.78-C3, 152.98-C2, 152.23-C7, 148.12-C12, 145.53-C13, 138.10-C4, 131.51-C8, 123.62-C15, 121.61-C11, 114.44-C14, 110.93-C10, 106.38-C6, 90.87-C1, 73.42-C16, 72.95-C21, 69.09-C26, 69.07-C31, 29.86-C17, 29.75-C22, 28.80-C27, 28.68-C32, 28.17-C18, 28.1-C19, 28.11-C23, 28.04-C24, 22.56-C28, 22.44-C29, 22.42-C33, 22.40-C34, 14.06 (×2)-C20 and C25, 13.98 (×2)-C30 and C35; HRMS calcd for C_35_H_51_O_8_ [M+H]^+^ 599.3578 found 599.3583.

**5-hydroxy-2-(3-hydroxy-4-(pent-4-en-1-yloxy)phenyl)-3**,**6**,**7-tris(pent-4-en-1-yloxy)-4H-chromen-4-one (4f)**. Yellow oil (36.9%). ^1^H NMR (CDCl_3_, 400 MHz) δ: 12.59 (s, 1H-7), 7.70-7.67 (m, 2H-15 and 11), 6.94 (d, J = 8.36 Hz, 1H-14), 6.47 (s, 1H-1), 5.91-5.84 (m, 4H-19, 24, 29 and 34), 5.13-4.97 (m, 8H-20, 25, 30 and 35), 4.16 (t, J = 6.44 Hz, 2H-16), 4.09-4.01 (m, 6H-21, 26 and 31), 2.32-2.28 (m, 6H-17, 22 and 27), 2.19-2.17 (m, 2H-32), 2.03-1.98 (m, 4H-18 and 23), 1.88-1.82 (m, 4H-28 and 33); ^13^C NMR (CDCl_3_, 100 MHz) δ: 179.01-C5, 158.44-C9, 155.78-C3, 153.02-C2, 152.24-C7, 148.05-C12, 145.58-C13, 138.35 (×2)-C19 and C4, 137.95-C24, 137.44-C29, 137.36-C34, 123.69-C8, 114.92-C15, 114.73-C14, 114.53-C11, 111.05-C10, 106.45-C6, 90.93-C1, 72.73-C16, 72.28-C21, 68.40-C26, 68.26-C31, 30.17-C17, 30.11-C22, 30.07-C27, 30.04-C32, 29.41-C18, 29.29-C23, 28.17-C28, 28.12-C33; HRMS calcd for C_35_H_43_O_8_ [M+H]^+^ 591.2952 found 591.2958.

**5-hydroxy-2-(3-hydroxy-4-((3-methylbut-2-en-1-yl)oxy)phenyl)-3**,**6**,**7-tris((3-methylbut-2-en-1-yl)oxy)-4H-chromen-4-one (4g)**. Yellow oil (34.8%). ^1^H NMR (CDCl_3_, 400 MHz) δ: 12.64 (s, 1H-7), 7.75-7.70 (m, 2H-15 and 11), 6.95 (d, J = 8.56 Hz, 1H-14), 6.48 (s, 1H-1), 5.63-5.43 (m, 4H-17, 22, 27 and 32), 1.83 (d, J = 4.36 Hz, 8H-16, 21, 26 and 31), 1.79 (s, 6H-19 and 20), 1.78 (s, 6H-24 and 25), 1.76 (s, 3H-29), 1.70 (s, 6H-30 and 34), 1.65 (s, 3H-35); ^13^C NMR (CDCl_3_, 100 MHz) δ: 179.17-C5, 158.46-C9, 156.01-C3, 153.21-C2, 152.25-C4, 147.91-C13, 145.68-C12, 139.43-C18, 139.33-C23, 138.55-C28, 137.55-C33, 131.12-C8, 121.64-C15, 120.64-C17, 119.85-C22, 118.96-C27, 118.77-C32, 114.48-C14, 111.37-C10, 106.32-C6, 91.19-C1, 69.21-C16, 68.87-C21, 65.96-C26, 65.82-C31, 25.81-C19, 25.78 (×2)C24 and C25, 25.76-C20, 18.31-C29, 18.26-C30, 17.98-C35, 17.88-C34; HRMS calcd for C_35_H_43_O_8_ [M+H]^+^ 591.2952 found 591.2949.

### 4.3. Biological Evaluation

#### 4.3.1. In Vitro Cytotoxicity Assay

Chronic myelogenous leukemia cell line K562, imatinib-resistant chronic myelogenous leukemia cell line K562R, human lymphocytic leukemia cell line U937, and human acute myeloid leukemia cell line KG-1 were used to evaluate the antineoplastic activities against leukemia of various novel compounds. These four are representative human leukemia cells that have diverse cell phenotypes and are distinct in biological relevance. All cells were cultured in RPMI 1640 (Gibco, Thermo Fisher Scientific, Walthman, MA, USA). All media were supplemented with 10% fetal bovine serum (FSP500, ExCell Bio, Shanghai, China) and 1% penicillin/streptomycin (03-031-1B, BI).

The effects of quercetagetin derivatives and vehicle control on cell viability were evaluated using the Cell Counting Kit 8 (CCK8, CK04, Dojindo Laboratories, Kumamoto, Japan) and all experiments were repeated three times [40].

#### 4.3.2. Apoptosis Analysis

K562 cells were seeded in a 6-well plate and grown overnight in the culture medium. After 24 h, cells were treated with different concentrations of compound **2a** for 48 h. After incubation, cells were collected and washed twice with cold PBS. Then, the cells were resuspended in 100 μL of 1× BD binding buffer solution (no. 556454, BD, New Jersey, USA) and stained with 7-ADD (no. 559925, BD) and annexin V (no. 556422, BD) in the dark for 15 min. Finally, 400 μL of 1× BD binding buffer solution was added to stop dyeing. The cells were then measured on a Guava easyCyte flow cytometer (Merck, Darmstadt, Germany) [41].

#### 4.3.3. Cell Cycle Analysis

Cells were seeded in a six-well plate and grown overnight in the culture medium. Cells were then treated with the test compound at various concentrations for 24 h. The cell cycle was measured using a Cell Cycle Analysis Kit (Sizhengbai Biotech, Beijing, China) according to the manufacturer’s instructions. Briefly, cells were harvested after treatment, washed twice with pre-cooled PBS, and then fixed in 95% ethanol overnight, and the cell number was adjusted to 1.0 × 10^6^ cells/mL before PI staining. The deoxyribonucleic acid (DNA) content of stained samples was analyzed using a Guava easyCyte flow cytometer (Merck, Rahway, NJ, USA) [42]

#### 4.3.4. Western Blot Analysis

Cells were treated with various concentrations of compound **2a** for 24 h. Then, the cells were lysed using 1 × SDS sample lysis buffer (CST recommended) with protease and phosphatase inhibitors. Cell lysates were loaded and electrophoresed onto 8–12% SDS-PAGE gel, and then the separated proteins were transferred to a PVDF film. The film was blocked with 5% BSA (Sigma-Aldrich, St. Louis, MO, USA) in TBS solution containing 0.5% Tween-20 for 4 h at rt, and then incubated with the corresponding primary antibody (1:2000–1:500) overnight at 4 °C. After washing with TBST, the HRP-conjugated secondary antibody was incubated for 2 h. The protein signals were visualized using an ECL Western blotting Detection Kit (Thermo Scientific, Grand Island, NY, USA), and detected with the Amersham Imager 600 system (GE, Boston, MA, USA). Primary antibodies against Bcl-2(3498), Mcl-1(4572), p-STAT3(9145), p-Erk1/2(4370) and p-AKT (4060) were purchased from Cell Signaling Technology (Boston, MA, USA).

### 4.4. In Silico Evaluation

#### 4.4.1. Molecular Docking Studies

The molecular docking studies were conducted using the SchrÖdinger Maestro molecular Modelling software (Glide, SchrÖdinger, LLC, New York, NY, USA, 2023). The crystal structures of Bcl-2(PDB ID: 8U27) and Mcl-1(PDB ID: 5IEZ) were downloaded from the RSCB database. The raw file was prepared for the docking assessment by the PrepWizard module of Maestro. The missing chains were added automatically by Prime and the protonation state was calculated by PropKa at physiological pH. Grid generation of Maestro was used to determine the docking grid, which was centered on the co-crystallized ligand and extended to a space of 25 × 25 × 25 Angstrom. Compound **2a** was sketched and cleaned in Maestro workspace and was prepared with energy minimization using an OPLS_2005 force field at physiological pH using the LigPrep module. The obtained ligands were subjected to Glide/XP docking protocols [43].

#### 4.4.2. ADME Studies and ‘BOILED-Egg’ Model Studies

For the ‘BOILED-egg’ model analysis and ADME studies, we utilized the ‘SwissADME’ webserver (http://www.swissadme.ch/index.php; accessed on: 24 October 2024) with the ‘SMILES’ strings of all compounds.

## 5. Conclusions

In this investigation, we designed and synthesized a series of quercetagetin derivatives and evaluated their antiproliferative activities against four leukemia cell lines (U937, K562, K562R and KG-1). Compound **2a** exhibits significant and potent anticancer activity against four leukemia cell lines (IC_50_ = 0.276, 0.159, 0.312, and 0.271 µM, respectively). Further, the flow cytometry assay indicated that **2a** can induce apoptosis in K562 cells and arrest the G2/M phase of the cell cycle effectively. In addition, mechanistic experiments showed that **2a** suppresses the expression of Bcl-2 and Mcl-1, indicating that **2a** is a potential inhibitor of Bcl-2 and Mcl-1. Molecular docking studies also revealed the crucial interactions between **2a** and apoptotic proteins, corroborating the conclusions of in vitro experiments. Finally, in silico studies predicted various ADME characteristics of these active derivatives, and it was revealed that **2a** is a suitable drug candidate for future Bcl-2 and Mcl-1 inhibitor studies.

## Data Availability

The data presented in this study are available on request from the corresponding author.

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
