# Peer review of "Synthesis and Biological Evaluation of Quercetagetin Derivatives as the Inhibitors of Mcl-1 and Bcl-2 Against Leukemia"

_ijms, 2025, doi:10.3390/ijms26062727_

Round 1
Reviewer 1 Report
Comments and Suggestions for Authors
Li et al. synthesis quercetagetin derivatives and investigated their potential effects against leukemia and as Bcl-2 and Mcl-1 inhibitors. This is an interesting paper, however, revisions are needed to improve the manuscript.
1. In Figures 7 and 8, only representative flow cytometric plots were shown, how about the corresponding bar graphs?
2. In Figure 9, the protein expressions of total Erk1/2, Stat3 and Akt should also be investigated.
3. Discussion should be written separately to discuss the current findings and compare these findings with other relevant studies.
4. More binding experiments are needed to prove whether compound 2a can be a potential compound as Bcl-2 and Mcl-1 inhibitors.
Comments on the Quality of English LanguageEnglish language needs to be improved by a native English speaker or English editing service.
Author Response
Reviewer #1
Comment 1: In Figures 7 and 8,only representative flow cytometric plots were shown, how about thecorresponding bar graphs?
Response: Thank you for your advice, we have updated the corresponding bar charts
Comment 2: In Figure 9, the protein expressions of total Erk1/2, Stat3 and Akt should also be2investigated
Response: We added the study and discussion of other related proteins based on our current experimental results
Comment 3: Discussion should be written separately to discuss the curent findings and compare thesefindings with other relevant studies
Response: Thank your reminder, we have adjusted the paper.
Comment 4: More binding experiments are needed to prove whether compound 2a can be a potential compound as Bcl-2 and Mcl-1 inhibitors.
Response: We agree with your comment that more experiments are needed to verify the results, but we are sorry to say that due to the limited financial support and time constraints we are unable to carry out more in-depth research, so we hope you will understand.

Reviewer 2 Report
Comments and Suggestions for Authors The manuscript entitled “Synthesis and biological evaluation of quercetagetin derivatives as the inhibitors of Mcl-1 and Bcl-2 against leukemia” (Manuscript ID: ijms-3436934) by Cheng et al., describes the synthesis of quercetagetin based new molecules coupled with different amino acids. These molecules were evaluated as Mcl-1 and Bcl-2 inhibitors by using human cancer cell lines. The following concerns need to be addressed before making a decision on the manuscript.1. Authors need to provide NMR Spectra for all the new compounds. The authors provided the 2D NMR spectral details in the manuscript. It will be useful for the reader if the spectra are included.
2. Terms like Very urgent (in the abstract), Capitalization of first words (2nd paragraph in the introduction), and words like synthesis instead of prepared need to be modified in the entire manuscript
3. The activity of the entire series coupled with amino acids resulted in low activity. What is the rationale for choosing amino acid coupling with Quercetagetin?
4. What interactions did authors observe with co-crystal in the molecular docking using 8U27 and provide details on retained or newly observed interactions in the active pocket?
5. All the figures and tables need Legends with details of n = number of experiments, details of the cell line, calculations, etc. Table 2, last column-rule of five violations instead of rule of five. Change Figure 3 to Scheme 1
6. Did the authors compare the activity of the compounds in a normal human cell line?
7. In Annexin assay, IC50 of compound 2a is 0.159 µM but the live cell population at 4 µM concentration is >84% which doesn’t match. The authors should also add control and compare the results. Further, including the concentration of DMSO used it is surprising to see 97% live cells in the presence of the DMSO.
8. Cell cycle analysis-G2/M, not G2 phase arrest.
9. Language polishing is suggested.
Author Response
Reviewer #2
Comment 1: Authors need to provide NMR Spectra for all the new compounds. The authors provided the2D NMR spectral details in the manuscript. lt will be useful for the reader if the spectra are
included.
Response: Yes we have provided NMR spectra for all compounds, the relevant images are collected in the supplementary material.
Comment 2: Terms like Very urgent (in the abstract), Capitalization of first words (2nd paragraph in theintroduction), and words like synthesis instead of prepared need to be modified in the entiremanuscrpt
Response: We apologise for the glitch, thanks for the heads up, we've got them corrected!
Comment 3: The activity of the entire series coupled with amino acids resulted in low activity. What is therationale for choosing amino acid coupling with Quercetagetin?
Response: Based on Kim's study(http://dx.doi.org/10.1016/j.bmc.2016.12.034), we believe that the introduction of amino acids may be able to enhance the reversal multi-drug resistance activity of flavonoids combined with methylation may yield derivatives with both anticancer and reversal multi-drug resistance activity, and then unfortunately none of the compounds in this class showed significant activity.
Comment 4: What interactions did authors observe with co-crystal in the molecular docking using 8U27 and provide details on retained or newly observed interactions in the active pocket?
Response: There are H-bonding interactions, π-π stacking and hydrophobic interactions between compound 2a and the co-crystal structures of 8U27. Methylation of quercetagetin resulted in a hydrophobic interaction between them.
Comment 5: All the figures and tables need Legends with details of n = number of experiments, details of the cell line, calculations, etc, Table 2, last column-rule of five violations instead of rule of five.Change Figure 3 to Scheme 1
Response: We have adjusted the content according to your suggestions.
Comment 6: Did the authors compare the activity of the compounds in a normal human cell line?
Response: We are sorry that we did not evaluate the toxicity of the compounds to normal human cells in this study.
Comment 7: In Annexin assay, IC50 of compound 2a is 0.159 µM but the live cell population at 4 µM concentration is >84% which doesn't match. The authors should also add control and comparethe results. Further, including the concentration of DMSO used it is surprising to see 97% live cells in the presence of the DMSO
Response: We apologise that we have not expressed the information clearly, we have added the concentration of DMSO. In the apoptosis analysis, we have reprocessed the data so that the results are more in line with the cytotoxicity assay, it is also possible that the anticancer activity of compound 2a is mediated through multiple mechanisms such as pyroptosis, which could lead to a poor match between the flow cytometry results and the cytotoxicity results.
Comment 8: Cell cycle analysis-G2/M, not G2 phase arrest
Response: We apologise for this error and thank you for the reminder, we have corrected the relevant content.
Comment 9: Language polishing is suggested
Response: Thanks for your advise, we have optimised our descriptions.

Round 2
Reviewer 1 Report
Comments and Suggestions for Authors
The authors have revised according to the comments, however, revisions are still needed to improve the manuscript.
1. WB data should be presented as p-Erk/Erk, p-Stat3/Stat3, p-Akt/Akt not as p-Erk, p-Stat3, p-Akt, respectively.
2. A conclusion paragraph should be added at the end of the discussion section.
3. Discussion needs to be revised by comparing and contrasting relevant studies of quercetagetin and its derivatives.
Author Response
Reviewer #1
Comment 1: WB data should be presented as p-Erk/Erk, p-Stat3/Stat3, p-Akt/Akt not as p-Erk, p-Stat3, p-Akt, respectively.
Response: Thank you for your guidance, we have modified the relevant content according to your requirements
Comment 2: A conclusion paragraph should be added at the end of the discussion section.
Response: Thank you for your suggestion, we have revised the content as required。
Comment 3: Discussion needs to be revised by comparing and contrasting relevant studies of quercetagetin and its derivatives.
Response: Thank you for your comment, we only found a handful of papers on quercetagetin derivatives and compared them.

Reviewer 2 Report
Comments and Suggestions for Authors
Comment 7: In Annexin assay, IC50 of compound 2a is 0.159 µM but the live cell population at 4 µM concentration is >84% which doesn't match. The authors should also add control and comparethe results. Further, including the concentration of DMSO used it is surprising to see 97% live cells in the presence of the DMSO
Response: We apologise that we have not expressed the information clearly, we have added the concentration of DMSO. In the apoptosis analysis, we have reprocessed the data so that the results are more in line with the cytotoxicity assay, it is also possible that the anticancer activity of compound 2a is mediated through multiple mechanisms such as pyroptosis, which could lead to a poor match between the flow cytometry results and the cytotoxicity results.
Comment: Pyroptosis can also be measured by Annexing V assay. Authors need to confirm whether the IC50 value for compound 2a in K562 cells is 0.1 µmol / L?!
Comments on the Quality of English LanguageNone
Author Response
Reviewer #2
Comment 1: Pyroptosis can also be measured by Annexing V assay. Authors need to confirm whether the IC50 value for compound 2a in K562 cells is 0.1 µmol / L?!
Response: Thank you for the reminder that we rechecked the data and repeated the cytotoxicity assay to determine the iC50 value for compound 2a, and it is possible that the difference in the results is due to the fact that we measured cytotoxicity with a processing time of 72 h, whereas flow cytometry was performed with a processing time of 48 h.

Round 3
Reviewer 2 Report
Comments and Suggestions for Authors
In conclusion, the mo- lecular docking studies explained the reasons why compound 2a showed bonding capac- ity to Bcl-2 and Mcl-1, finally leading to high inhibitory activities. This statement appears exaggerated. Instead the molecular docking studies provided the insights on compound 2a binding capacity to Bcl-2 and Mcl-1. would suffice.
Comments on the Quality of English LanguageNone
Author Response
Comment 1: In conclusion, the molecular docking studies explained the reasons why
compound 2a showed bonding capacity to Bcl-2 and Mcl-1, finally leading to high
inhibitory activities. This statement appears exaggerated. Instead the molecular
docking studies provided the insights on compound 2a binding capacity to Bcl-2 and
Mcl-1. would suffice.
Response:Thank you for your suggestion, we agree that this is a bit of an
overstatement, so we have adjusted the expression.
